# Healthcare Priorities, Barriers, and Preferences According to a Community Health Needs Assessment in Jazan, Saudi Arabia: A Cross-Sectional Study

**DOI:** 10.3390/healthcare13020107

**Published:** 2025-01-08

**Authors:** Ibrahim M. Gosadi

**Affiliations:** Department of Family and Community Medicine, Faculty of Medicine, Jazan University, P.O. Box 2349, Jazan 82621, Saudi Arabia; igosadi@jazanu.edu.sa

**Keywords:** health priorities, barriers, perception, preference, Jazan, Saudi Arabia

## Abstract

**Background**: Community needs assessments can provide valuable insights concerning the health of communities. This study aims to measure health priorities according to community members in Jazan, Saudi Arabia, to assess healthcare service utilization, barriers to accessing these services, and preferences for utilizing government or private healthcare services. **Method**: Multistage sampling was utilized to reach a sample of adults in community settings. The assessment was performed via personal interviews utilizing a structured questionnaire to measure demographics, health priorities, service utilization, barriers, and preferences for healthcare settings. Chi-squared tests, Student’s *t*-tests, and multivariate regression analysis were used to assess the differences between demographics and service utilization according to the preferred healthcare settings. **Results**: A total of 3411 participants were recruited for the assessment. The mean age of the participants was 34 years, and 51% were male. The participants viewed diabetes as the most important health condition. Emergency services and primary care were viewed as the most important healthcare services, and time constraints were the main barriers to healthcare accessibility. Thirty-six percent of the participants preferred to utilize healthcare services in the private sector, where age, gender, nationality, education, income, housing type, and family size were statistically associated with the preference for seeking healthcare in either the private or government sectors (*p* < 0.05). **Conclusions**: Future assessment is required to recruit healthcare providers and decision-makers to understand the process of strengthening multidisciplinary collaboration to tackle chronic diseases such as diabetes, strengthen the role of emergency and primary healthcare services, and address time constraints pertaining to healthcare accessibility.

## 1. Introduction

Community needs assessments can provide valuable insights concerning the health status of individuals in a community, determine risk factors and causes of diseases, and identify suitable healthcare services and programs [1]. The growing burden of chronic diseases and the aging population impact the planning and delivery of healthcare services. Saudi is one of the countries affected by the rise in chronic, non-communicable diseases, such as diabetes, hypertension, dyslipidemia, and obesity [2]. Saudi Vision 2030 involves major transformations and emphasizes the enhancement of integration within the health sector, comprehensiveness, and the effectiveness of healthcare delivery while prioritizing disease prevention and access to healthcare [3].

The burden of illnesses has witnessed a shift from communicable diseases to non-communicable diseases in Saudi Arabia during the last 40 years. This shift is associated with the implementation of several schemes, such as vaccination programs, to control communicable diseases. However, the variability of the geography and socioeconomic status of different regions in the country can leave some people more exposed to certain illnesses and risk factors in comparison to others. For example, the Jazan region in the south of Saudi Arabia has been found to be more vulnerable to vector-borne diseases in comparison to other regions in the country [4].

One of the early health needs assessment projects was conducted in the Jazan region in 2006. The most important findings of this assessment were related to the perception of a limited number of healthcare providers, problems related to environmental health and communicable diseases, and concerns about chronic diseases, such as hypertension, diabetes, asthma, and joint diseases [5]. After 17 years, a study was conducted in 2023 targeting a sample of 3083 affiliates of Jazan University, which is the only university in the region. The study found that diabetes was viewed as the most important healthcare issue, and the limited availability of appointments was the main barrier to healthcare access [6]. This suggests a periodic change in disease perception and healthcare priorities in Jazan during the past two decades and a shift from viewing infectious diseases as the most important to non-communicable diseases.

Although diabetes and other non-communicable diseases are perceived as important in the Jazan region according to recent reports, it is possible to argue that risk factors associated with chronic non-communicable diseases are relatively prevalent and difficult to control. This notion is supported by recent studies that measured healthcare-seeking behavior and lifestyle determinants associated with chronic non-communicable diseases. For example, in a study that recruited 1957 students from Jazan University, nearly 25% were either overweight or obese, and the majority of the students exhibited inadequate dietary habits and low levels of physical activity [7]. Furthermore, a study that targeted adults in the Jazan region investigated the utilization of routine medical checkup services for the early detection and follow-up of chronic non-communicable diseases, and the results revealed low utilization of these services, indicating access barriers and limited awareness [8].

Community health can vary according to the socioeconomic characteristics of the communities, epidemiology of diseases, perception of what constitutes a health need, and availability and accessibility of healthcare services. It is possible to argue that needs can vary between communities and countries based on the variations in these elements. For example, in a US community needs assessment, it was noted that cardiovascular diseases, and spine and joint diseases were identified as main health issues, while barriers to health were more associated with weight, physical activity, dietary concerns, and occupational concerns [9]. In another community health needs assessment conducted in Israel, it was noted that health needs can vary according to ethnicity in the same community where Arab residents reported factors such as discrimination and domestic voidance as perceived determinants of health, while Jew residents reported different factors such as accessibility to mental health services and transportations as determinants affecting their health [10].

Other situational health needs assessments can be related to specific diseases, political circumstances, or disaster situations. An example of a health needs assessment pertaining to a specific disease is the assessment of the needs of patients with cardiovascular diseases in Nepal [11]. Targeting a specific group according to vulnerability, such as health needs assessment among refugees in the US is an example of targeting a community with unique political circumstances [12]. Assessment of health needs of communities after a natural disaster such as the assessment of community health needs after Hurricane Irma in the US is an example of how health needs can be affected by disaster situations [13]. This suggests the importance of conducting health needs assessment within communities and considering the variability of needs according to the circumstances of the local communities.

Current evidence suggests a higher burden associated with a rise in the prevalence of chronic non-communicable diseases in the Jazan region. The rise in specific conditions and associated risk factors might be due to the overall adherence and preferences concerning the utilization of preventive and curative healthcare services [14,15,16]. This indicates the importance of performing periodic community health needs assessments to measure perceptions in communities regarding the most important health concerns and the healthcare services required to meet healthcare needs. This study aims to measure health priorities according to community members in the Jazan region. Furthermore, the study also aims to assess healthcare service utilization, barriers to the use of these services, and preferences regarding the utilization of government or private healthcare services.

## 2. Materials and Methods

### 2.1. Study Context and Settings

For this study, a community health needs assessment was carried out between May and June 2024. The assessment was performed in the region of Jazan in southwest Saudi Arabia, which comprises 17 administrative areas and has a total population of nearly 1.4 million. The region has several geographic features, including coastal areas, islands, and mountains [17]. These variations are likely to impact the delivery of healthcare services and access to healthcare. The study included adults living in the region regardless of their nationality. Individuals who were not residing in the region at the time of recruitment or participants under the age of 18 were excluded. Ethical approval to conduct the assessment was granted via the Standing Committee for Scientific Research of Jazan University (approval number REC-45/05/848 dated 26 November 2023). The study was performed in accordance with the Declaration of Helsinki, and informed consent was obtained from the participants before their recruitment.

### 2.2. Data Collection Tool

A structured questionnaire was utilized for data collection and was completed during the interviews with the participants. The questionnaire was developed based on the literature associated with performing community health needs assessments [1,18]. The content validity of the questionnaire was assessed by a panel of specialists in community medicine to ensure its comprehensiveness in measuring the health needs of the community in the Jazan region.

The questionnaire comprised six main components. The first section collected the participants’ demographic data, the second section collected information on their family history of disease and diagnosed conditions, the third section measured the participants’ healthcare-seeking behavior, the fourth section inquired about their perception of the most important healthcare issues in the region, the fifth section assessed the participants’ perception of the most important healthcare services that should be provided for the community, and the sixth section inquired about the participants’ perception of the most important healthcare delivery barriers. The data collection tool was produced in both the Arabic and English languages to meet the language preference of the targeted community, which is mainly native Arabic speakers.

### 2.3. Data Collection Process

Data collection was performed in community settings within the region, and trained medical students performed the interviews in the participants’ households. The training involved sessions to familiarize the students with the assessment tool and the process of conducting the interviews to enhance consistency. Quality control for the interviews was performed during the community assessment via supervised debriefs and frequent interview checks.

The students were divided into twelve groups (six male and six female groups) to ensure the appropriate distribution of the data collection teams on assessment sites and to ensure equal representation of male and female community members. Each team was composed of 15–18 students. Nearly 200 students participated in the community assessments. The assessments were performed in households distributed in 17 governorates in the Jazan region. The assessments were ensured to measure the needs of individuals considering the geographical variations in the region where assessments were performed in urban and rural areas within coastal, islands, and mountain areas of the region. After the completion of the training session, the students were instructed to pilot the questionnaire on one participant to test the familiarity of the participants with the questionnaire items. The piloting was performed to ensure the face validity of the questionnaire and its ability to assess the health needs of the targeted community.

The sample size was estimated based on the following formula: nh = (deff) (Z2) (P) (1 − P) (k)/(Ã)(d2). A total of 1600 household surveys was estimated, assuming that the z-statistic equals 1.96 for a 95% confidence level. The sample design effect (deff) was set at 2. A proportion of 20% was used for the non-response multiplier (k). The indicator (P) was set at 50% to provide the largest sample size to allow the assessment of multiple health conditions and healthcare-seeking behavior. Within each family, a minimum of one parent and one adult offspring were invited to participate in the study to enhance the generalizability of the results to wider age groups.

### 2.4. Data Analysis

The data analysis was performed using R software (version 4.2.3). Proportions and frequencies were utilized to summarize binary and categorical data, and means and standard deviations were used to summarize continuous data. Normal distribution of continuous data was performed via visualization of the data and assessment for the presence of skewness. A chi-squared test or Student’s *t*-test was used to measure the variations in demographic characteristics concerning the preference for using government or private sector healthcare services. Multivariate regression analysis was additionally performed to provide an assessment of the influence of sociodemographic factors on the preference of selecting private over governmental healthcare services. A *p*-value of 0.05 or less was designated as statistically significant for the applied statistical tests.

## 3. Results

A total of 3411 participants were recruited from various governances in the region. The mean age of the participants was 34 years (SD = 15), and the distribution of the sample according to gender was nearly equal (49% female and 51% male). When the participants were asked about their preferred location for receiving healthcare services, 2193 (64%) reported a preference for utilizing government services, while 1218 (36%) indicated a preference for using private sector healthcare services.

Table 1 displays the sample characteristics classified according to the preference for using government or private healthcare services. Statistically significant differences were noted in the characteristics of the sample according to the preferred setting for healthcare services according to age, gender, nationality, education level, income, housing type, and family size. The frequency of participants who preferred to use private healthcare services was higher among female, non-Saudi participants with higher education levels, higher incomes, who owned bigger housing units (villas), and who belonged to families with a smaller number of children (*p*-value < 0.05). Table 2 shows the findings of the multivariate regression analysis where higher odds of utilizing private healthcare services were noted among females, those who were non-Saudi, those with a monthly income of more than 15,000 SAR, those who owned villas, and families with a smaller number of children (*p* values < 0.05). This suggests that the preference for using private healthcare services is mainly influenced by gender and economic status.

The family history of morbidities and causes of death are displayed in Table 3. The most frequently reported family history disease was diabetes, followed by hypertension. Additionally, the most frequently reported cause of death among the sample was related to road traffic accidents, followed by death due to cardiovascular diseases and cancer. Healthcare-seeking behavior according to the settings did not statistically differ according to family history of diseases nor the cause of death among family members except for family history of death due to road traffic accidents. Those with a family history of death due to road traffic accidents reported a preference for seeking healthcare in a governmental setting in comparison to private healthcare settings (*p* value < 0.05).

Table 4 displays the distribution of the participants according to diagnosed medical conditions. The most frequently diagnosed conditions among the sample are dental conditions, followed by physical injury. Additionally, the most frequently diagnosed chronic diseases were hypertension, diabetes, asthma, and obesity. Classification of the sample according to the preferred healthcare settings shows a lack of statistically significant difference in preferred healthcare settings according to the history of diagnosed condition except for being diagnosed with either hypertension, diabetes, dengue fever, or AIDS. Those who reported being diagnosed with hypertension or diabetes indicated a high preference to seek healthcare services in governmental settings (*p* values < 0.01), while those who were diagnosed with dengue fever or AIDS showed a higher preference to seek healthcare from the private sector (*p* values < 0.05). Although the study did not investigate why those with dengue fever or AIDS may show a higher preference to seek healthcare in private settings, it is possible to argue that the indicated preference might be associated with the urgency of the condition and the need to receive healthcare in a rapid manner, as in the case of dengue fever, or to receive healthcare in a more convenient and private setting, which may be the case with private healthcare settings.

When the participants were asked about the frequency of utilizing healthcare services, it was noted that seeking lifestyle counseling and performing routine medical checkups were less frequently used in comparison to services provided by dentists or physicians. The healthcare-seeking behavior among the participants according to the healthcare setting is illustrated in Table 5. There was no statistically significant difference concerning location for seeking dental or physician healthcare services. However, routine medical checkups and lifestyle counseling visits were more likely to be performed in government healthcare settings in comparison to private healthcare settings (*p*-value < 0.001).

Figure 1 displays the perception of the participants regarding the most important health conditions in the community. Diabetes was considered the most important condition (39%), followed by cancer (13%), and hypertension (12%). When the participants were asked about their opinion of the most important healthcare services that should be provided to the community (see Figure 2), the most frequently indicated were emergency services (23%), followed by primary healthcare services (20%), and specialist healthcare services (14%). Finally, Figure 3 displays the perceptions of the participants regarding the most important barriers to obtaining healthcare. The most frequently mentioned barriers were related to the availability of appointments (28%), followed by time limitations and prolonged waiting times (20%), and the cost of healthcare (10%).

## 4. Discussion

The current community assessment indicates a higher reliance on government healthcare services, which can be mainly explained by the availability of free healthcare services for Saudi nationals. However, as more than one-third of the sample preferred utilizing private-sector healthcare services, this indicates a shift toward the private sector among people with certain socioeconomic characteristics. When the participants were asked about their family history of medical conditions, diabetes and hypertension were most frequently reported, and road traffic accidents and cardiovascular diseases were the most frequently reported causes of death among family members. The most frequently diagnosed conditions among the sample were metabolic diseases, such as hypertension, diabetes, obesity, and asthma.

The high prevalence of metabolic diseases may have aided in creating a perception among the community concerning diabetes as the most important health issue in the Jazan region. Additionally, physical injuries were reported as one of the most frequently diagnosed conditions among the sample. These perceptions and reported frequencies may explain the reason for perceiving primary healthcare services and emergency services as the most important. Furthermore, when the participants were asked about the most important healthcare barriers concerning their receipt of healthcare services, the limited availability of appointments was the most frequently reported. This may partially explain the perception among community members concerning the importance of specialist care that is needed for more complex conditions that can be associated with the prevalence of metabolic diseases.

The findings of the current investigation can be compared to similar investigations. In a previous study that involved a sample of university affiliates in the Jazan region, diabetes was viewed as the most important health condition, followed by cancer, hypertension, and obesity [6]. Perceiving diabetes as the most important health condition in the community stems from the increasing incidence of the disease in the country. In a national study that assessed the burden of diseases in Saudi Arabia between 2010 and 2017 involving more than 24,000 households, it was concluded that diabetes, hypertension, cardiovascular diseases, and cancer were the most prevalent conditions in the country [19]. Furthermore, in a study that reviewed historical prevalence rates of diabetes during the last 20 years in the Gulf Council countries, including Saudi Arabia, it was noted that the prevalence of diabetes nearly doubled between 2000 and 2019 (from 9% to 18%) [20]. In response, the Saudi Ministry of Health has carried out national surveys to identify the burden of chronic conditions and the prevalence of the associated risk factors [21,22] and introduced an awareness platform promoting a healthy lifestyle as a prevention measure for chronic non-communicable diseases [23].

The current findings indicate that nearly half of the sample (48%) are either overweight or obese, while 11% are underweight. Similar studies that assessed BMI profiles among younger populations in the Jazan region such as schools and university students indicated different findings. In a study by Alqassimi et al. which recruited a sample of 228 medical students, it was concluded that the prevalence of overweight and obesity was 28.3%, while 17.3% were underweight [24]. In another investigation that targeted 1957 university students from the Jazan region, it was similarly reported that the prevalence of overweight and obesity represented 25.45%, while those who are classified as underweight constituted 21% of the sample [7]. Finally, in a study that recruited 570 school students in the Jazan region, the prevalence of students who were classified as overweight and obese reached 22% while those classified as underweight represented 20% of the sample [25]. The findings of the current study and similar investigations that involved younger samples suggest that older individuals are more likely to be affected by a higher prevalence of overweight and obesity while younger individuals can be more subjected to having underweight status. This indicates the importance of considering variation in BMI category variation when implementing healthy body weight initiatives according to the age groups.

The current sample indicated that emergency services are viewed as the most important healthcare service, followed by primary healthcare and specialist care. The high demand for emergency services can be partially explained by the higher prevalence of physical injuries and diseases that might require urgent care in vulnerable groups, such as elderly people with diabetes. These findings are consistent with a report by Alferdaus and Shaher, who investigated the trauma system in Saudi Arabia and indicated that, in 2018, among the nearly 12,000 cases transferred via ambulance services in the Jazan region, 46% were due to injuries, with the majority caused by road traffic accidents [26]. Nonetheless, the incidence of road traffic accidents in Saudi Arabia decreased by 38% between 2016 and 2022 according to the Saudi Ministry of Health [27]. However, the response from the community suggests the need to strengthen the emergency services in the region. This is supported by a review that assessed the utilization of emergency services in Saudi Arabia, where the rise in chronic metabolic diseases has led to an increased rate of emergency department visits and increased duration of emergency department stays [28].

In addition to emergency services, primary healthcare, and specialist care were also viewed as important services. This is supported by a review conducted by Al Asmri et al., which assessed the transformation of the primary care system in Saudi Arabia and indicated that the transition should focus on meeting the burden associated with the increased prevalence of chronic diseases in Saudi communities [29]. In a review by Al Khashan et al. that assessed the primary healthcare services reform in Saudi Arabia, it was concluded that the reform, which was initiated in 2016 as a part of the Saudi Vision 2030, has resulted in the increased utilization of primary healthcare services in the country. Nonetheless, Al Khashan et al. indicated that more effort should be put into enhancing human resources in primary care centers and strengthening inter-sectorial collaboration [30]. This is augmented by the possibility of difficulties concerning the referral system between primary healthcare services and specialist care in the Jazan region that might influence patients’ overall satisfaction with the healthcare services provided [31].

According to the current sample, dental conditions were reported to be the most frequently experienced. Nonetheless, when the participants were asked about their frequency of visiting a dentist, only 14% reported visiting a dentist every six months, and the majority reported either visiting a dentist when suffering from a dental complaint or never visiting one. This indicates the low utilization of routine dental services in the community. This finding is similar to that of Sahab et al., who measured the utilization of dental services among adults in Saudi Arabia. They concluded that symptomatic attendance was the main reason for visiting a dentist and more emphasis should be placed on increasing attendance to prevent dental issues [32].

The preference for healthcare services among the current sample indicates a higher utilization of routine medical checkups and lifestyle counseling in the government sector, with a statistically significant difference. However, visits to dentists and physicians were not statistically different when comparing between government and private sector services. Although government services are freely available to Saudi citizens [33], observing a shift toward seeking healthcare services from the private sector indicates the presence of barriers to accessing services in the government sector. This may be partially explained by the finding of the current investigation that the most frequently reported barrier to receiving a healthcare service was the limited availability of appointments.

Studies that assessed barriers or facilitators of accessing healthcare services in other international contexts indicate that accessibility can be influenced by the operational healthcare system, socioeconomic factors, and cultural factors. In an Australian study that compared healthcare utilization among Arabic-speaking migrants and Caucasian English-speaking patients with type two diabetes, it was concluded that English-speaking patients were more likely to seek healthcare and have higher accessibility, while Arabic-speaking migrants tend to delay access due to social, cultural, religious beliefs, and unfamiliarity of available healthcare services [34]. In a systematic review that investigated healthcare accessibility among indigenous adolescents from Australia, Canada, New Zealand, and the USA, it was concluded that fear, lack of culturally suitable services, and confidentiality were common barriers reported by the adolescents and their parents [35]. This is contrary to the findings of the current study where factors related to awareness about healthcare services, social factors such as fear of stigma, and language barriers were not commonly reported by the current sample and more emphasis was given to other factors such as availability of appointments and cost of healthcare.

The findings of the current study indicate that the nature of the healthcare system can impose barriers to healthcare accessibility. This is similar to a US study that involved a sample of 2194 adult participants where the most important barriers were related to healthcare system barriers, such as limited appointments, healthcare system navigation, cost of healthcare, and discrimination barriers, such as barriers based on gender, ability to pay, and ethnicity [36]. The current study did not assess discrimination barriers in a Saudi Arabian context. However, the healthcare system barriers identified in the US study are similar to the findings of the current study.

The current assessment has multiple strengths and weaknesses. One of the main strengths is the relatively representative sample size of the community with various demographic characteristics and, thus, may provide an accurate assessment of the health needs among the community and appropriate generalizability. The main limitation of the current assessment is its reliance on the community members to provide a viewpoint of their main health needs without resorting to healthcare providers and decision-makers, which can be an area for future investigation. Additionally, the study did not assess the quality of healthcare services in the region which can be recommended for future research.

## 5. Conclusions

The findings of this study indicate that diabetes is perceived as the most important health condition. Emergency healthcare, primary healthcare, and specialist care were viewed as the most important services. The main healthcare limitations were related to prolonged waiting times to access services. More than a third of the sample reported a preference for using private healthcare services, where economic status seems to influence the preference for selecting private healthcare services over government healthcare services. via observing that availability of appointments is the main reported healthcare barrier in the current sample and via observing that economic status is an important predictor of utilizing private healthcare services, it is possible to argue that those with higher income status are more likely to choose to overcome the difficulty of accessing healthcare services associated with appointments availability and prolonged waiting time via consulting healthcare services in the private section.

The findings of the current study indicate the need for further assessments that aim to recruit healthcare providers and decision-makers. These assessments are needed to understand the process of strengthening multidisciplinary collaboration to tackle chronic diseases such as diabetes, strengthen the role of the emergency and primary healthcare service providers, and to address time constraints pertaining to healthcare accessibility. The findings of the assessments should be followed up by the development of policies and initiatives to enhance public health interventions aiming to reduce the burden of chronic non-communicable diseases in the region and to enhance the accessibility of healthcare services. Additionally, future research should focus on the assessment of the association between the perception of quality of healthcare services according to the healthcare settings.

## Figures and Tables

**Figure 1 healthcare-13-00107-f001:**
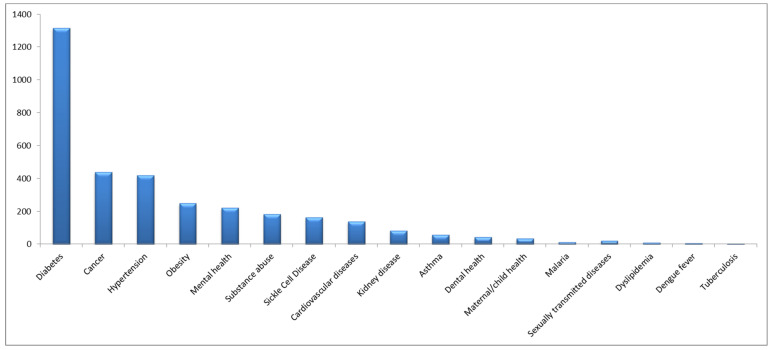
Frequencies of reported perceptions of the most important health issues among 3411 participants from the Jazan region of Saudi Arabia.

**Figure 2 healthcare-13-00107-f002:**
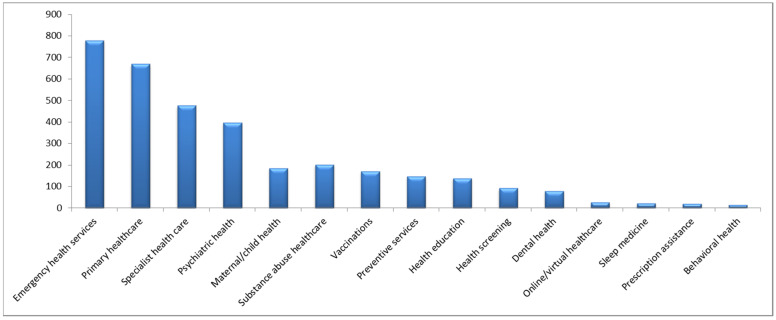
Frequencies of reported perceptions of the most important healthcare services among 3411 participants from the Jazan region of Saudi Arabia.

**Figure 3 healthcare-13-00107-f003:**
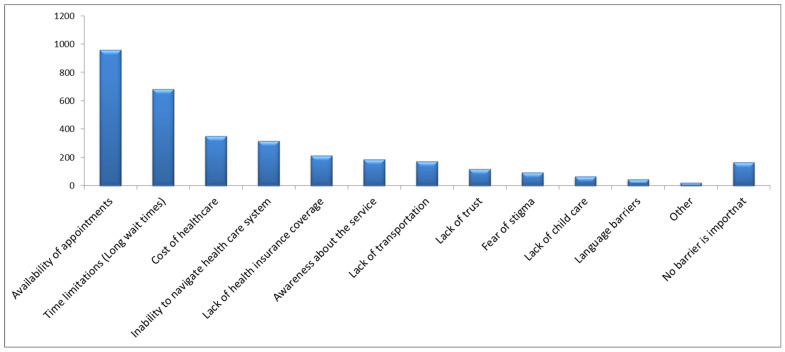
Frequencies of reported perceptions of the most important barriers to accessing healthcare services among 3411 participants from the Jazan region of Saudi Arabia.

**Table 1 healthcare-13-00107-t001:** Sample characteristics according to the utilization of government or private healthcare services among 3411 participants in the Jazan region of Saudi Arabia.

Variable	Governmental		Private		*p* Value *
	Frequency/Mean	Proportion/SD	Frequency/Mean	Proportion/SD	
Age	34	15	33	13	0.039
Gender					0.001
Female	1031	47%	645	53%	
Male	1162	53%	573	47%	
Nationality					0.01
Saudi	2142	98%	1170	96%	
Non-Saudi	51	2%	48	4%	
Education					0.008
Illiterate	67	3%	20	2%	
Elementary	71	3%	25	2%	
Intermediate	74	3%	45	4%	
Secondary	751	34%	384	32%	
University	1156	53%	696	57%	
Postgraduate	74	3%	48	4%	
Marital status					0.081
Single	1087	50%	591	49%	
Married	1006	46%	585	48%	
Divorced	44	2%	26	2%	
Widowed	56	3%	16	1%	
Monthly income (SAR)					0.004
≤5000	726	33%	357	29%	
>5000–≤10,000	430	20%	209	17%	
>10,000–≤15,000	482	22%	265	22%	
>15,000	555	25%	387	32%	
Residence area					0.056
Rural	1376	63%	723	59%	
Urban	817	37%	495	41%	
Housing type					2.121 × 10^−6^
Owned villa	950	43%	617	51%	
Owned apartment	440	20%	240	20%	
Owned traditional house	531	24%	205	17%	
Rented	272	12%	156	13%	
Family size	7.1	2.9	6.7	2.8	0.0001
BMI					0.786
Underweight	243	11%	136	11%	
Normal weight	907	41%	501	41%	
Overweight	654	30%	373	31%	
Obese	389	18%	208	17%	
Smoking					0.801
Second hand smoker	107	5%	62	5%	
Current smoker	215	10%	119	10%	
Ex-smoker	102	5%	48	4%	
Never smoker	1769	81%	989	81%	

* *p*-values were obtained from the chi-squared test for categorical data and Student’s *t*-test for continuous variables. SAR: Saudi Arabian riyal. BMI: body mass index

**Table 2 healthcare-13-00107-t002:** Multivariate regression analysis of utilization of government or private healthcare services among 3411 participants in the Jazan region of Saudi Arabia.

	Odds of Utilizing Private Healthcare Services		
Predictors	Odds Ratios	CI	*p*
(Intercept)	1.37	0.60–3.06	0.449
Age	0.99	0.98–0.99	0.043
Gender			
Female (reference)			
Male	0.73	0.63–0.86	<0.001
Nationality			
Non-Saudi (reference)			
Saudi	0.46	0.29–0.71	0.001
Education			
Illiterate (reference)			
Elementary	1.00	0.50–2.01	0.990
Intermediate	1.43	0.74–2.82	0.291
Secondary	1.10	0.62–2.01	0.753
University	1.26	0.72–2.28	0.434
Postgraduate	1.15	0.58–2.30	0.698
Social status			
Single (reference)			
Divorced	1.16	0.66–2.01	0.591
Married	1.14	0.89–1.46	0.310
Widowed	0.68	0.34–1.31	0.260
Income (SAR)			
≤5000 (reference)			
>5000–≤10,000	1.04	0.83–1.29	0.754
>10,000–≤15,000	1.10	0.89–1.36	0.386
>15,000	1.40	1.14–1.73	0.001
Residence			
Rural (reference)			
Urban	1.05	0.90–1.22	0.552
Housing type			
Rented (reference)			
Owned apartment	1.04	0.80–1.37	0.765
Owned traditional house	0.86	0.65–1.15	0.312
Owned villa	1.30	1.02–1.67	0.038
Family size	0.96	0.94–0.97	0.008
BMI	1.00	0.99–1.02	0.584
Smoking			
Never (reference)			
Second hand smoker	1.04	0.75–1.44	0.795
Current smoker	1.12	0.87–1.44	0.375
Ex-smoker	1.05	0.72–1.51	0.807

SAR: Saudi Arabian Riyals, BMI: body mass index.

**Table 3 healthcare-13-00107-t003:** Family history and causes of death of family members according to 3411 participants from the Jazan region of Saudi Arabia classified according to their utilization of government or private healthcare services.

Variable	Governmental		Private		*p* Value *
	Frequency	Proportion	Frequency	Proportion	
Family history of diabetes	1140	52%	630	52%	0.912
Family history of hypertension	1121	51%	603	50%	0.387
Family history of cardiovascular disease	303	14%	204	17%	0.024
Family history of cancer	213	10%	133	11%	0.289
Family history of dyslipidemia	123	6%	69	6%	0.972
Death among family due to road traffic accident	337	15%	150	12%	0.016
Death among family due to cardiovascular disease	224	10%	146	12%	0.124
Death among family due to cancer	238	11%	141	12%	0.556

* *p*-values were obtained from the chi-squared test.

**Table 4 healthcare-13-00107-t004:** Prevalence of diagnosed medical conditions among 3411 participants from the Jazan region of Saudi Arabia classified according to the utilization of government or private healthcare services.

Diagnosed Condition	Governmental		Private		*p* Value *
	Frequency	Proportion	Frequency	Proportion	
Dental condition	327	15%	199	16%	0.290
Physical injury	318	15%	182	15%	0.764
Hypertension	326	15%	118	10%	2.11 × 10^−5^
Diabetes	315	14%	109	9%	5.66 × 10^−6^
Asthma	151	7%	86	7%	0.902
Obesity	131	6%	68	6%	0.696
Dyslipidemia	70	3%	42	3%	0.762
Dengue fever	67	3%	62	5%	0.003
Mental condition	59	3%	44	4%	0.160
Malaria	68	3%	28	2%	0.211
Sickle cell disease	65	3%	21	2%	0.035
Cardiovascular disease	55	3%	19	2%	0.089
Kidney disease	33	2%	8	1%	0.044
Hepatitis B	26	1%	9	1%	0.287
Tuberculosis	14	1%	5	<1%	0.537
Cancer	10	<1%	8	1%	0.596
AIDS	7	<1%	11	1%	0.0445
Hepatitis C	11	1%	5	<1%	0.911
Thalassemia	10	0%	5	0%	0.991

* *p*-values were obtained from the chi-squared test.

**Table 5 healthcare-13-00107-t005:** Healthcare-seeking behavior among 3411 participants from the Jazan region of Saudi Arabia classified according to the preferred healthcare facility.

Variables	Governmental		Private		*p* Value *
	Frequency	Proportion	Frequency	Proportion	
Dentist visits					0.640
Minimum of once every 6 months	316	14%	166	14%	
Once a year	216	10%	134	11%	
Only when suffering from a dental condition	1217	55%	682	56%	
Rarely or never	444	20%	236	19%	
Physician visits					0.0812
Minimum of once every 6 months	307	14%	138	11%	
Once a year	116	5%	55	5%	
Only when suffering from a medical condition	1540	70%	900	74%	
Rarely or never	230	10%	125	10%	
Routine medical checkups					0.0032
Minimum of once every 6 months	438	20%	213	17%	
Once a year	209	10%	148	12%	
Only when referred by a health professional	878	40%	442	36%	
Rarely or never	668	30%	415	34%	
Lifestyle counseling visits					5.95 × 10^−5^
Minimum of once every 6 months	210	10%	98	8%	
Once a year	139	6%	68	6%	
Only when referred by a healthcare professional	503	23%	211	17%	
Rarely or never	1341	61%	841	69%	

* Chi-squared tests.

## Data Availability

The raw data supporting the conclusions of this article are available upon reasonable request to the corresponding author.

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
