# Peer review of "Healthcare Priorities, Barriers, and Preferences According to a Community Health Needs Assessment in Jazan, Saudi Arabia: A Cross-Sectional Study"

_healthcare, 2025, doi:10.3390/healthcare13020107_

Round 1
Reviewer 1 Report
Comments and Suggestions for Authors
Dr. Ibrahim M. Gosadi presents a study examining health priorities, barriers to healthcare, and preferences for government or private healthcare services in the Jazan region of Saudi Arabia. While the manuscript addresses important issues, there are several areas where constructive improvements could enhance its rigor, clarity, and relevance.
Major Comments
1. The title inaccurately refers to Saudi Arabia as a whole, while the study is confined to the Jazan region. The title should reflect the regional scope and focus on Jazan.
2. The scale of the study, which involved data collection from over 3,400 participants, demands contributions from a team. The authro has informed in methodology that medial students have collected the data. The lack of co-authors or acknowledgments for roles such as data collection, analysis, and manuscript preparation raise concerns about transparency and ethical authorship. The author should clarify and acknowledge the roles of all contributors.
3. The manuscript relies on references authored by Dr. Gosadi, which undermines objectivity. While referencing one’s prior work is acceptable, this over-reliance limits engagement with broader research. The author should replace self-citations with independent studies unless they are directly relevant.
4. The manuscript does not provide sufficient details about how the questionnaire was validated or piloted. The sampling strategy lacks clarity, particularly how it ensured representation across diverse subpopulations within Jazan. The study uses chi-squared tests and t-tests to examine differences in healthcare preferences and barriers, but these basic tests limit the ability to account for confounding factors. Multivariable regression models could provide a more comprehensive understanding of how variables such as age, income, and education independently influence healthcare preferences. For example, logistic regression could assess predictors of preference for private versus government healthcare services. Figures and tables require better captions to explain their significance and how they relate to the study objectives.
5. The discussion does not sufficiently explore why diabetes is seen as the most critical health issue or propose actionable solutions for barriers like time constraints and appointment availability. The findings would benefit from comparisons with other regions or countries with similar challenges, providing a broader context and relevance.
6. Ethical transparency on participant confidentiality should be addressed along with attaching the survey questionairre with appropriate references as supplementary.
Author Response
Comment: Dr. Ibrahim M. Gosadi presents a study examining health priorities, barriers to healthcare, and preferences for government or private healthcare services in the Jazan region of Saudi Arabia. While the manuscript addresses important issues, there are several areas where constructive improvements could enhance its rigor, clarity, and relevance.
Response: The author appreciates the supportive comment of the reviewer. We confirm that several modifications were applied to the revised manuscript to improve its writing quality.
Major Comments
Comment: 1. The title inaccurately refers to Saudi Arabia as a whole, while the study is confined to the Jazan region. The title should reflect the regional scope and focus on Jazan.
Response: The title has been modified to accurately refer to the region. The title is now modified and highlighted with yellow in the revised manuscript as the following:
‘Healthcare priorities, barriers, and preferences according to a community health needs assessment in Jazan, Saudi Arabia: a cross-sectional study’
Comment: 2. The scale of the study, which involved data collection from over 3,400 participants, demands contributions from a team. The authro has informed in methodology that medial students have collected the data. The lack of co-authors or acknowledgments for roles such as data collection, analysis, and manuscript preparation raise concerns about transparency and ethical authorship. The author should clarify and acknowledge the roles of all contributors.
Response: The author agrees with the comment of the reviewer. As explained in the methodology, this manuscript is related to a community-based project to assess health needs of communities in the Jazan region of Saudi Arabia and is a result of collaboration between teaching staff in the department of Family and Community Medicine, and students in the faculty of Medicine, Jazan University, in Jazan region. To enhance the scientific transparency, the acknowledgment is now modified as the following and highlighted with yellow in the revised manuscript to acknowledge the contribution of the department, data analysis specialist, and the students:
‘Acknowledgments: The authors acknowledges the contribution of the Head of Department of Family and Community Medicine, Dr.Ahmad Alqassim in providing the administrative support for conducting the community health needs assessment. The author also acknowledges the assistance of Dr.Mohammad Jareebi in conducting statistical analysis via R software. Finally, the author acknowledges the contribution of the medical students in performing the data collection.’
Comment: 3. The manuscript relies on references authored by Dr. Gosadi, which undermines objectivity. While referencing one’s prior work is acceptable, this over-reliance limits engagement with broader research. The author should replace self-citations with independent studies unless they are directly relevant.
Response: The author of the manuscript agrees with the comment of the reviewer. We confirm that the manuscript has six references authored by Gosadi IM and colleagues. Nonetheless, the six references are highly related to the current manuscript and provides scientific background concerning the community needs and healthcare accessibility in the region. Therefore, we confirm that these references are directly relevant to the current manuscript and avoiding them would negatively affect the scientific objectivity of the current assessment.
Comment: 4. The manuscript does not provide sufficient details about how the questionnaire was validated or piloted.
Response: Further details are now provided concerning piloting and validation of the questionnaire in the revised manuscript as the following:
‘A structured questionnaire was utilized for data collection and was completed during the interviews with the participants. The questionnaire was developed based on literature associated with performing community health needs assessments [1, 18]. The content validity of the questionnaire was assessed by a panel of specialists in community medicine to ensure its comprehensiveness to measure health needs of the community in Jazan region.’
‘Data collection was performed in community settings within the region, and trained medical students performed the interviews in the participants’ households. The training involved sessions to familiarize the students with the assessment tool and the process of conducting the interviews to enhance consistency. Quality control for the interviews was performed during the community assessment via supervised debriefs and frequent interview checks.’
‘After completion of the training session, the students were instructed to pilot the questionnaire on one participants to tests the familiarity of the participants with the questionnaire items. The piloting was performed to ensure the face validity of the questionnaire and its ability to assess the health needs of the targeted community.’
Comment: The sampling strategy lacks clarity, particularly how it ensured representation across diverse subpopulations within Jazan.
Response: Further details are now added to indicate how the sampling was performed to ensure appropriate assessment of the subpopulations with the region. The following text was added to the revised methodology and highlighted with yellow:
‘The students were divided into 12 groups ( six male and six female groups) to ensure appropriate distribution of the data collection teams on assessment sites and to ensure equal representation of male and female community members. Each team was compromised of 15-18 students. Nearly 200 students participated in the community assessments. The assessments were performed in households distributed in 17 governorates in the Jazan region. The assessments were ensured to measure the needs of individuals considering the geographical variations of the region where assessments were performed in urban and rural areas within coastal, islands, and mountain areas of the region.’
Comment: The study uses chi-squared tests and t-tests to examine differences in healthcare preferences and barriers, but these basic tests limit the ability to account for confounding factors. Multivariable regression models could provide a more comprehensive understanding of how variables such as age, income, and education independently influence healthcare preferences. For example, logistic regression could assess predictors of preference for private versus government healthcare services.
Response: as per the request of the reviewer, further analysis utilizing multivariate regression analysis was performed. Changes in the manuscript in response to this comment involved modifying the abstract, the methodology section, and the results section, adding a new table (table 2), and modifying the numbering of the tables accordingly. Sections modified in the abstract, methodology, results, and added table are highlighted with yellow in the revised manuscript to indicate areas of change.
Comment: Figures and tables require better captions to explain their significance and how they relate to the study objectives.
Response: An effort was made to make the titles of the tables and figures clear to explain the nature of the data, its settings, and its relation to the study objectives. Additionally, Statistically significant values are indicated based on the performed statistical tests within the table and the relevant P values are presented accordingly.
Comment: 5. The discussion does not sufficiently explore why diabetes is seen as the most critical health issue or propose actionable solutions for barriers like time constraints and appointment availability. The findings would benefit from comparisons with other regions or countries with similar challenges, providing a broader context and relevance.
Response: Further discussion is now added to clarify why diabetes is viewed as an important condition by the local community and comparison of challenges identified in the current sample to other countries with broader context. The following is discussing those notions in the revised manuscript and is highlighted with yellow to indicate areas of change as the following:
‘The high prevalence of metabolic diseases may have aided in creating a perception among the community concerning diabetes as the most important health issue in the Jazan region.’
‘[6]. Perceiving diabetes as the most important health condition in the community stems from the increasing incidence of the disease in the country. In a national study that assessed the burden of diseases in Saudi Arabia between 2010 and 2017 involving more than 24,000 households, it was concluded that diabetes, hypertension, cardiovascular diseases, and cancer were the most prevalent conditions in the country [19]. Furthermore, in a study that reviewed historical prevalence rates of diabetes during the last 20 years in the Gulf Council countries, including Saudi Arabia, it was noted that the prevalence of diabetes nearly doubled between 2000 and 2019 (from 9% to 18%) [20]. In response, the Saudi Ministry of Health has carried out national surveys to identify the burden of chronic conditions and the prevalence of the associated risk factors [21, 22] and introduced an awareness platform promoting a healthy lifestyle as a prevention measure for chronic non-communicable diseases [23].’
‘Although government services are freely available to Saudi citizens [33], observing a shift toward seeking healthcare services from the private sector indicates the presence of barriers to accessing services in the government sector. This may be partially explained by the finding of the current investigation that the most frequently reported barrier to receiving a healthcare service was the limited availability of appointments.
Studies that assessed barriers or facilitators of accessing healthcare services in other international contexts indicates that accessibility can be influenced by the operational healthcare system, socioeconomic factors, and cultural factors. In Australian study that compared healthcare utilization among Arabic-speaking migrants and Caucasian Eng-lish-speaking patients with type two diabetes, it was concluded that English-speaking patients were more likely to seek healthcare and have higher accessibility, while Ara-bic-speaking migrants tend to delay access due to social, cultural, religious beliefs, and unfamiliarity of available healthcare services [34]. In systematic review that investigated healthcare accessibility among indigenous adolescents from Australia, Canada, New Zealand, and USA, it was concluded that fear, lack of culturally suitable services, and confidentiality were common barriers reported by the adolescents and their parents [35]. This is contrary to the findings of the current study where factors related to awareness about healthcare services, social factors such as fear of stigma, and language barriers were not commonly reported by the current sample and more emphasis was given for other factors such as availability of appointments and cost of healthcare.
The findings of the current study indicates that the nature of the healthcare system can imposed barriers to healthcare accessibility. This is similar to a US study that in-volved a sample of 2194 adult participants where most important barriers were related to healthcare system barriers, such as limited appointments, healthcare system navigation, cost of healthcare, and discrimination barriers, such as barriers based on gender, ability to pay, and ethnicity [36]. The current study did not assess discrimination barriers in a Saudi Arabian context. However, the healthcare system barriers identified in the US study are similar to the findings of the current study.’
Comment: 6. Ethical transparency on participant confidentiality should be addressed along with attaching the survey questionairre with appropriate references as supplementary.
Response: The author of the manuscript appreciates the comment of the reviewer. We declare the ethical transparency and participants confidentiality are ensured as declared in the study context section of the methodology of the revised manuscript. We also confirm that all variables and assessment items measured in the current analysis are provided within the tables and figures of the manuscript and this can allow replication of the study findings in other settings. Therefore, considering avoiding adding unnecessary supplementary materials, all assessment items are displayed within the manuscript accordingly.
Reviewer 2 Report
Comments and Suggestions for Authors
Thank you for inviting me to review the article titled “Healthcare priorities, barriers, and preferences according to a community health needs assessment in Saudi Arabia”. The authors addressed an essential public health issue. However, there are numerous concerns that need to be rectified or revised for better clarity.
Title: The title is misleading. Authors need to specify the study design.
In addition, I strongly suggest following the STROBE statement while presenting the manuscript.
Introduction: The author made a good attempt to convey the disease burden in KSA and Jazan region. However, the knowledge gap needs to be explained more. The transition between concepts is missing. It is good to see the author mention the local context (Vision 2030). But the other references are very old. Needs assessments are constantly changing, and authors need to update the latest (past 5 years) references.
Methods:
The manuscript significantly lacks methodological rigor. Kindly explain all the details of the STROBE statement. Attaching a separate checklist would be transparent.
Inclusion and exclusion criteria: Need to be clearly explained in the methods. How did you collect data from the child?
Data collection tool: Need extensive explanation as basic details are missing. The author(s) developed the tool based on existing literature (1, 13). They are from different settings, and the authors need to explain the development process and all validation in detail. Adding the data collection and EFA (validity) would be more transparent. What language did you use for data collection? How did you handle expatriates’ data collection? Another language?
Data analysis: Did the author make normality assumption tests to present data as mean and SD?. Considering that the large data and the possibility of confounding bias are very high, it is strongly recommended that a multivariate analysis be conducted to get a valid conclusion and policy changes.
Results:
So, the expatriates constitute only 2% of the study population. But the sampling method explanation, I was assuming that it is a probability sampling method. In that case, the proportion of expatriates should have been high in accordance with the Jazan/Saudi data.
Also, it is interesting to see 11% of participants are underweight in a country with a good economic situation.
Kindly make a multivariate analysis for the associated factors analysis.
Discussion:
The organization and structure of the discussion must be improved. Also, make it according to the revised findings.
Moreover, I suggest that rather than more generic policy implications, make it which is unique to the current research.
Comments on the Quality of English LanguageOverall, it is fine. Key areas for improvement include sentence structure, redundancy, and word choice.
Author Response
Comment: Thank you for inviting me to review the article titled “Healthcare priorities, barriers, and preferences according to a community health needs assessment in Saudi Arabia”. The authors addressed an essential public health issue. However, there are numerous concerns that need to be rectified or revised for better clarity.
Response: : The author appreciates the supportive comment of the reviewer. We confirm that several modifications were applied to the revised manuscript to improve its writing quality.
Comment: Title: The title is misleading. Authors need to specify the study design.
Response: The title has been modified to accurately refer to the study design. The title is now modified and highlighted with yellow in the revised manuscript as the following:
‘Healthcare priorities, barriers, and preferences according to a community health needs assessment in Jazan, Saudi Arabia: a cross-sectional study’
Comment: In addition, I strongly suggest following the STROBE statement while presenting the manuscript.
Response: An effort has been made to follow the STROBE statement for observational study while presenting the findings of the manuscript.
Comment: Introduction: The author made a good attempt to convey the disease burden in KSA and Jazan region. However, the knowledge gap needs to be explained more. The transition between concepts is missing. It is good to see the author mention the local context (Vision 2030). But the other references are very old. Needs assessments are constantly changing, and authors need to update the latest (past 5 years) references.
Response: The author agrees with the comment of the reviewer. More explanation is now added to fill gaps in the knowledge and to explain various concepts associated with performing community health needs assessments in different international context. Additionally, more recent references are now added to the introduction. The following is now added to the introduction and highlighted with yellow in the revised manuscript:
‘Community health can vary according to socioeconomic characteristics of the communities, epidemiology of diseases, perception of what constitute a health need, and availability and accessibility of healthcare services. It is possible to argue that needs can vary between communities and countries based on the variations of these elements. For example, in a US community needs assessment, it was noted that cardiovascular diseases, and spine and joint diseases were identified as main health issues, while barriers to health were more associated with weight, physical activity, dietary concerns, and occupational concerns [9]. In another community health needs assessment conducted in Israel, it was noted that health needs can vary according to the ethnicity in the same community where Arab residents reported factors such as discrimination and domestic voidance as perceived determinants of health, while Jew residents reported different factors such as accessibility to mental health services and transportations as determinants affecting their health [10].
Other situational health needs assessment can be related to specific diseases, political circumstances, or disaster situations. An example of health needs assessment pertaining to a specific disease is the assessment of needs of patients with cardiovascular diseases in Nepal [11]. Targeting a specific group according to vulnerability, such as health needs assessment among refugees in the US is an example of targeting a community with unique political circumstances [12]. Assessment of health needs of communities after a natural disaster such as assessment of community health needs after Hurricane Irma in the US is an example of how health needs can be affected by disaster situation [13]. This suggests the importance of conducting health needs assessment within communities and to consider variability of needs according to the circumstances of the local communities.’
Methods:
Comment: The manuscript significantly lacks methodological rigor. Kindly explain all the details of the STROBE statement. Attaching a separate checklist would be transparent.
Response: More details are now added to the methodology. Modified sections of the methodology are highlighted with yellow in the revised manuscript. Additionally, STROBE statement checklist is now completed and attached for reference.
Comment: Inclusion and exclusion criteria: Need to be clearly explained in the methods.
Response: Inclusion and exclusion criteria are now detailed in the materials and methods section of the revised manuscript and highlighted with yellow as the following:
‘The study included adults living in the region regardless of their nationality. Individuals whom were not residing in the region at the time of recruitment or participants under the age of 18 were excluded.’
Comment: How did you collect data from the child?
Response: We confirm that all participants recruited in the current investigation were adults. No underage participants were recruited in the current investigation. However, as the sampling unit in the current assessment was household, one parent and one adult offspring were targeted. This is indicated in the materials and methods section in the revised manuscript as the following:
‘Within each family, a minimum of one parent and one adult offspring were invited to participate in the study to enhance the generalizability of the results to wider age groups.’
Comment: Data collection tool: Need extensive explanation as basic details are missing. The author(s) developed the tool based on existing literature (1, 13). They are from different settings, and the authors need to explain the development process and all validation in detail. Adding the data collection and EFA (validity) would be more transparent. What language did you use for data collection? How did you handle expatriates’ data collection? Another language?
Response: Further details are now provided concerning questionnaire language, piloting and validation of the questionnaire in the revised manuscript as the following:
‘A structured questionnaire was utilized for data collection and was completed during the interviews with the participants. The questionnaire was developed based on literature associated with performing community health needs assessments [1, 18]. The content validity of the questionnaire was assessed by a panel of specialists in community medicine to ensure its comprehensiveness to measure health needs of the community in Jazan region.
The questionnaire was comprised of six main components. The first section collected the participants’ demographic data, the second section collected information on their family history of disease and diagnosed conditions, the third section measured the participants’ healthcare-seeking behavior, the fourth section inquired about their perception of the most important healthcare issues in the region, the fifth section assessed the participants’ perception of the most important healthcare services that should be provided for the community, and the sixth section inquired about the participants’ perception of the most important healthcare delivery barriers. The data collection tool was produced in both Arabic and English languages to meet the language preference of the targeted community which is mainly native Arabic speakers.’
‘After completion of the training session, the students were instructed to pilot the questionnaire on one participants to tests the familiarity of the participants with the questionnaire items. The piloting was performed to ensure the face validity of the questionnaire and its ability to assess the health needs of the targeted community.’
Comment: Data analysis: Did the author make normality assumption tests to present data as mean and SD?.
Response: Distribution of continuous data was tested via visualization of continuous data to detect the presence of any skewness. After performing the visualization via histogram and testing the normal curve. Section indicating testing the normal distribution of the data is now added and highlighted with yellow in the revised manuscript as the following:
‘Normal distribution of continuous data was performed via visualization of the data and assessment for presence of skewness.’
Comment: Considering that the large data and the possibility of confounding bias are very high, it is strongly recommended that a multivariate analysis be conducted to get a valid conclusion and policy changes.
Response: as per the request of the reviewer, further analysis utilizing multivariate regression analysis was performed. Changes in the manuscript in response to this comment involved modifying the abstract, the methodology section, and the results section, adding a new table (table 2), and modifying the numbering of the tables accordingly. Sections modified in the abstract, methodology, results, and added table are highlighted with yellow in the revised manuscript to indicate areas of change. We also confirm that the findings of the multivariate analysis supports the findings of the study.
Results:
Comment: So, the expatriates constitute only 2% of the study population. But the sampling method explanation, I was assuming that it is a probability sampling method. In that case, the proportion of expatriates should have been high in accordance with the Jazan/Saudi data.
Response: We agree with the comment of the reviewer that expatriates constitute higher proportion than 2% in the Saudi community. However, it must be noted that majority of expatriates in the Jazan region are male workers whom are more likely to be living in male-only residential compounds and are less likely to live in Saudi Arabia with their families. This style of living of expatriates is common in high economic Gulf Region such as Saudi Arabia, Qatar, and the UAE which have high demand for expatriate workers. The current investigation targeted household in the Jazan region, and therefore, expatriates whom are not living in households with their families were less likely to be recruited. This explains the small representation of expatriates in the current sample.
Comment: Also, it is interesting to see 11% of participants are underweight in a country with a good economic situation.
Response: The current sample identified a prevalence of 11% of underweight. To further explore this notion, similar investigations that assess BMI profile in the region were consulted. It was noted that younger individuals were more likely to exhibit higher prevalence of underweight status in comparison to older subjects. A further discussion is now added to report the prevalence of underweight status in the current sample and how it compares to similar research. The following was added to the discussion section and highlighted with yellow in the revised manuscript:
‘The current findings indicates that nearly half of the sample (48%) are either over-weight or obese, while 11% are underweight. Similar studies that assessed BMI profiles among younger populations in the Jazan region such as schools and university students indicated different findings. In a study by Alqassimi et al. which recruited a sample of 228 medical students, it was concluded that prevalence of overweight and obesity was 28.3%, while 17%.3% were underweight [24]. In another investigation that targeted 1957 university students from Jazan region, it was similarly reported that prevalence of overweight and obesity represented 25.45%, while those whom are classified as under-weight constituted 21% of the sample [7]. Finally, in a study that recruited 570 school students in the Jazan region, the prevalence of students whom were classified as over-weight and obese reached 22% while those classified as underweight represented 20% of the sample [25]. The findings of the current study and similar investigations that involved younger samples suggests that older individuals are more likely to be affected by higher prevalence of overweight and obesity while younger individuals can be more subjected to having underweight status. This indicates the importance of considering variation of BMI category variation when implementing healthy body weight initiatives according to the age groups.’
Comment: Kindly make a multivariate analysis for the associated factors analysis.
Response: as per the request of the reviewer, further analysis utilizing multivariate regression analysis was performed. Changes in the manuscript in response to this comment involved modifying the abstract, the methodology section, and the results section, adding a new table (table 2), and modifying the numbering of the tables accordingly. Sections modified in the abstract, methodology, results, and added table are highlighted with yellow in the revised manuscript to indicate areas of change. We also confirm that the findings of the multivariate analysis supports the findings of the study.
Discussion:
Comment: The organization and structure of the discussion must be improved. Also, make it according to the revised findings.
Response: The author appreciates the comment of the reviewer. The discussion is now revised to improve the writing quality. More sections were added to ensure the discussion is made according to the current findings. The modified sections of the discussion are highlighted with yellow in the revised manuscript.
Comment: Moreover, I suggest that rather than more generic policy implications, make it which is unique to the current research.
Response: More policy implications that are unique to the findings of the current study are now added to the conclusion section of the revised manuscript. The following was added and highlighted with yellow to the conclusion:
‘The findings of the current study indicate the need for further assessments that aim to recruit healthcare providers and decision makers. These assessment are needed to understand the process of strengthening multidisciplinary collaboration to tackle chronic diseases such as diabetes, strengthen the role of the emergency and primary healthcare service providers, and ti address time constraints pertaining to healthcare accessibility. The findings of the assessments should be followed up by development of policies and initiatives to enhance public health interventions aiming to reduce the burden of chronic non-communicable diseases in the region and to enhance accessibility of healthcare services. Additionally, future research should focus on assessment of association between perception of quality of healthcare services according to the healthcare settings.’
Comments on the Quality of English Language
Overall, it is fine. Key areas for improvement include sentence structure, redundancy, and word choice.
Response: a UK-based English language editing service was consulted to enhance the writing quality of the manuscript. We also confirm that further editing service can be implemented according to the journal’s instruction.

Reviewer 3 Report
Comments and Suggestions for Authors
A useful study which should be published.
The findings regarding health care seeking behaviour from either governmental healthcare services or private healthcare services is interesting and could be explored further.
In the “Results” section (lines 137 to 145, and Table 1)) you highlight the findings according to the preference for using government or private healthcare services. You also state “This suggests that the preference for using private healthcare services is mainly influenced by gender and economic status.”
In the “Discussion” section (lines 283-284), you indicate “a higher utilization of routine medical checkups and lifestyle counseling in the government sector, with a statistically significant difference.”
In lines 287 to 291, “Although government services are freely available to Saudi citizens [26], observing a shift toward seeking healthcare services from the private sector indicates the presence of barriers to accessing services in the government sector. This may be partially explained by the finding of the current investigation that the most frequently reported barrier to receiving a healthcare service was the limited availability of appointments.” It seems you are suggesting that obtaining an appointment is the reason for seeking care in the private sector (despite the additional cost to the patient).
Yet, in the “Conclusion” section (lines 303 – 306), you state “More than a third of the sample reported a preference for using private healthcare services, where economic status seems to influence the preference for selecting private healthcare services over government healthcare services.” This implies that patients with wealth/higher income can afford to side-step the government services to gain access to the private sector. This could be more explicitly stated.
Throughout the text you state using private healthcare services is influenced by gender, economic status, as well as availability of appointments. This could be made more consistent and tied together in the “Conclusion”.
Also, have you looked at perceptions of quality of services? It could be that the population perceives private sector healthcare services as being of better quality. If you have not addressed quality of services, this should be mentioned as a limitation to the study.
Author Response
Comment: A useful study which should be published.
Response: The author appreciates the supportive comment of the reviewer.
Comment: The findings regarding health care seeking behaviour from either governmental healthcare services or private healthcare services is interesting and could be explored further.
Response: Tables 2 and 3 are now presented with classifying the sample according to the type of healthcare (governmental or private). Furthermore, additional analysis utilizing multivariate regression analysis was performed to investigate factors associated with the preference of utilizing governmental or private healthcare services. Changes in the manuscript in response to this comment involved modifying the abstract, the methodology section, and the results section, adding a new table (table 2), and modifying the numbering of the tables accordingly. Sections modified in the abstract, methodology, results, and added table are highlighted with yellow in the revised manuscript to indicate areas of change.
Comment: In the “Results” section (lines 137 to 145, and Table 1)) you highlight the findings according to the preference for using government or private healthcare services. You also state “This suggests that the preference for using private healthcare services is mainly influenced by gender and economic status.”
In the “Discussion” section (lines 283-284), you indicate “a higher utilization of routine medical checkups and lifestyle counseling in the government sector, with a statistically significant difference.” In lines 287 to 291, “Although government services are freely available to Saudi citizens [26], observing a shift toward seeking healthcare services from the private sector indicates the presence of barriers to accessing services in the government sector. This may be partially explained by the finding of the current investigation that the most frequently reported barrier to receiving a healthcare service was the limited availability of appointments.” It seems you are suggesting that obtaining an appointment is the reason for seeking care in the private sector (despite the additional cost to the patient).Yet, in the “Conclusion” section (lines 303 – 306), you state “More than a third of the sample reported a preference for using private healthcare services, where economic status seems to influence the preference for selecting private healthcare services over government healthcare services.” This implies that patients with wealth/higher income can afford to side-step the government services to gain access to the private sector. This could be more explicitly stated.
Throughout the text you state using private healthcare services is influenced by gender, economic status, as well as availability of appointments. This could be made more consistent and tied together in the “Conclusion”.
Response: The author of the manuscript agrees with the comment of the reviewer. The following is added to the conclusion section and highlighted with yellow in the revised manuscript to indicate how economic status can influence preference of seeking healthcare in the private sector to overcome accessibility issues:
‘The main healthcare limitations were related to prolonged waiting times to access services. More than a third of the sample reported a preference for using private healthcare services, where economic status seems to influence the preference for selecting private healthcare services over government healthcare services. Via observing that availability of appointments is the main reported healthcare barrier in the current sample and via observing that economic status is an important predictor of utilizing private healthcare services, it is possible to argue that those with higher income status are more likely choose to overcome the difficulty of accessing healthcare services associated with appointments availability and prolonged waiting time via consulting healthcare services in the private section.’
Comment: Also, have you looked at perceptions of quality of services? It could be that the population perceives private sector healthcare services as being of better quality. If you have not addressed quality of services, this should be mentioned as a limitation to the study.
Response: Unfortunately, the current study did not investigate perception of quality of life. This is now indicated as a limitation and an area for further research. Sections modified and highlighted with yellow in the revised manuscript to indicate this notion are as the following:
‘The main limitation of the current assessment is its reliance on the community members to provide a viewpoint of their main health needs without resorting to healthcare providers and decision makers, which can be an area for future investigation. Additionally, the study did not assess quality of healthcare services in the region which can be recommended for future research.’
‘Additionally, future research should focus on assessment of association between perception of quality of healthcare services according to the healthcare settings.’
Round 2
Reviewer 1 Report
Comments and Suggestions for Authors
The author has made commendable efforts in revising the manuscript to address the reviewer's comments effectively. owever, to further strengthen the manuscript’s scientific rigor and compliance with reporting standards, the inclusion of a STROBE statement as appendix is recommended.
Author Response
Comment: The author has made commendable efforts in revising the manuscript to address the reviewer's comments effectively. owever, to further strengthen the manuscript’s scientific rigor and compliance with reporting standards, the inclusion of a STROBE statement as appendix is recommended.
Response: We appreciate the supportive comment of the reviewer. The STROBE statement is attached with the resubmission as recommended.
Reviewer 2 Report
Comments and Suggestions for Authors
Dear authors,
thanks for making great efforts in revising paper.
i have few more comments.
have you checked the word count of abstract ?
Also, it is suggested to keep exact p value rather than mentioning as <0.05 or > 0.05.
I believe author might have modified conclusion based on some other reviewer’s suggestions. But it should be short.
wish you all the best
Author Response
Comment: Dear authors,
thanks for making great efforts in revising paper.
i have few more comments.
Response: We appreciate the supportive comment of the reviewer. All comments in round two have been addressed accordingly. The modified sections of the manuscript are highlighted with yellow to indicate areas of change.
Comment: have you checked the word count of abstract ?
Response: Yes, the word count is currently 254 words. We acknowledge that the word count is exceeding the limit by four words (250) but this was due to a reviewer’s request of adding more details to the abstract.
Comment: Also, it is suggested to keep exact p value rather than mentioning as <0.05 or > 0.05.
Response: As per the reviewer request, the exact p values are now added to the tables of the revised manuscript. Modifications are highlighted with yellow to indicate application of changes to the relevant tables.
Comment: I believe author might have modified conclusion based on some other reviewer’s suggestions. But it should be short.
wish you all the best
Response: We appreciate the supportive comment of the reviewer. We agree that the conclusion should be short to convey summary of the findings and provide practical and research recommendation in a brief manner. However, the conclusion section was prolonged due to the reviewer’s comment requesting addition of some vital components.